# FLIPOUT: EFFICIENT PSEUDO-INDEPENDENT WEIGHT PERTURBATIONS ON MINI-BATCHES

**Yeming Wen, Paul Vicol, Jimmy Ba**
University of Toronto
Vector Institute
`wenyemin,pvicol,jba@cs.toronto.edu`

**Dustin Tran**
Columbia University
Google
`trandustin@google.com`

**Roger Grosse**
University of Toronto
Vector Institute
`rgrosse@cs.toronto.ca`

## ABSTRACT

Stochastic neural net weights are used in a variety of contexts, including regularization, Bayesian neural nets, exploration in reinforcement learning, and evolution strategies. Unfortunately, due to the large number of weights, all the examples in a mini-batch typically share the same weight perturbation, thereby limiting the variance reduction effect of large mini-batches. We introduce flipout, an efficient method for decorrelating the gradients within a mini-batch by implicitly sampling pseudo-independent weight perturbations for each example. Empirically, flipout achieves the ideal linear variance reduction for fully connected networks, convolutional networks, and RNNs. We find significant speedups in training neural networks with multiplicative Gaussian perturbations. We show that flipout is effective at regularizing LSTMs, and outperforms previous methods. Flipout also enables us to vectorize evolution strategies: in our experiments, a single GPU with flipout can handle the same throughput as at least $40$ CPU cores using existing methods, equivalent to a factor-of-$4$ cost reduction on Amazon Web Services.

## 1 INTRODUCTION

Stochasticity is a key component of many modern neural net architectures and training algorithms. The most widely used regularization methods are based on randomly perturbing a network's computations (Srivastava et al., 2014; Ioffe & Szegedy, 2015). Bayesian neural nets can be trained with variational inference by perturbing the weights (Graves, 2011; Blundell et al., 2015). Weight noise was found to aid exploration in reinforcement learning (Plappert et al., 2017; Fortunato et al., 2017). Evolution strategies (ES) minimizes a black-box objective by evaluating many weight perturbations in parallel, with impressive performance on robotic control tasks (Salimans et al., 2017).

Some methods perturb a network's activations (Srivastava et al., 2014; Ioffe & Szegedy, 2015), while others perturb its weights (Graves, 2011; Blundell et al., 2015; Plappert et al., 2017; Fortunato et al., 2017; Salimans et al., 2017). Stochastic weights are appealing in the context of regularization or exploration because they can be viewed as a form of posterior uncertainty about the parameters. However, compared with stochastic activations, they have a serious drawback: because a network typically has many more weights than units, it is very expensive to compute and store separate weight perturbations for every example in a mini-batch. Therefore, stochastic weight methods are typically done with a single sample per mini-batch. In contrast, activations are easy to sample independently for different training examples within a mini-batch. This allows the training algorithm to see orders of magnitude more perturbations in a given amount of time, and the variance of the stochastic gradients decays as $1/N$, where $N$ is the mini-batch size. We believe this is the main reason stochastic activations are far more prevalent than stochastic weights for neural net regularization. In other settings such as Bayesian neural nets and evolution strategies, one is forced to use weight perturbations and live with the resulting inefficiency.

In order to achieve the ideal $1/N$ variance reduction, the gradients within a mini-batch need not be independent, but merely uncorrelated. In this paper, we present flipout, an efficient method for decorrelating the gradients between different examples without biasing the gradient estimates. Flipout applies to any perturbation distribution that factorizes by weight and is symmetric around 0—including DropConnect, multiplicative Gaussian perturbations, evolution strategies, and variational Bayesian neural nets—and to many architectures, including fully connected nets, convolutional nets, and RNNs.

In Section 3, we show that flipout gives unbiased stochastic gradients, and discuss its efficient vectorized implementation which incurs only a factor-of-2 computational overhead compared with shared perturbations. We then analyze the asymptotics of gradient variance with and without flipout, demonstrating strictly reduced variance. In Section 4, we measure the variance reduction effects on a variety of architectures. Empirically, flipout gives the ideal $1/N$ variance reduction in all architectures we have investigated, just as if the perturbations were done fully independently for each training example. We demonstrate speedups in training time in a large batch regime. We also use flipout to regularize the recurrent connections in LSTMs, and show that it outperforms methods based on dropout. Finally, we use flipout to vectorize evolution strategies (Salimans et al., 2017), allowing a single GPU to handle the same throughput as 40 CPU cores using existing approaches; this corresponds to a factor-of-4 cost reduction on Amazon Web Services.

## 2 BACKGROUND

### 2.1 WEIGHT PERTURBATIONS

We use the term "weight perturbation" to refer to a class of methods which sample the weights of a neural network stochastically at training time. More precisely, let $f(x, W)$ denote the output of a network with weights $W$ on input $x$. The weights are sampled from a distribution $q_{\boldsymbol{\theta}}$ parameterized by $\boldsymbol{\theta}$. We aim to minimize the expected loss $\mathbb{E}_{(x,y)\sim\mathcal{D}, W\sim q_{\boldsymbol{\theta}}}[\mathcal{L}(f(x, W), y)]$, where $\mathcal{L}$ is a loss function, and $\mathcal{D}$ denotes the data distribution. The distribution $q_{\boldsymbol{\theta}}$ can often be described in terms of perturbations: $W = \overline{W} + \Delta W$, where $\overline{W}$ are the mean weights (typically represented explicitly as part of $\boldsymbol{\theta}$) and $\Delta W$ is a stochastic perturbation. We now give some specific examples of weight perturbations.

**Gaussian perturbations.** If the entries $\Delta W_{ij}$ are sampled independently from Gaussian distributions with variance $\sigma_{ij}^2$, this corresponds to the distribution $W_{ij} \sim \mathcal{N}(\overline{W}_{ij}, \sigma_{ij}^2)$. Using the reparameterization trick (Kingma & Welling, 2014), this can be rewritten as $W_{ij} = \overline{W}_{ij} + \sigma_{ij}\epsilon_{ij}$, where $\epsilon_{ij} \sim \mathcal{N}(0, 1)$; this representation allows the gradients to be computed using backprop. A variant of this is *multiplicative* Gaussian perturbation, where the perturbations are scaled according to the weights: $W_{ij} \sim \mathcal{N}(\overline{W}_{ij}, \sigma_{ij}^2\overline{W}_{ij}^2)$, or $W_{ij} = \overline{W}_{ij}(1 + \sigma_{ij}\epsilon_{ij})$, where again $\epsilon_{ij} \sim \mathcal{N}(0, 1)$. Multiplicative perturbations can be more effective than additive ones because the information content of the weights is the same regardless of their scale.

**DropConnect.** DropConnect (Wan et al., 2013) is a regularization method inspired by dropout (Srivastava et al., 2014) which randomly zeros out a random subset of the weights. In the case of a 50% drop rate, this can be thought of as a weight perturbation where $\overline{W} = W/2$ and each entry $\Delta W_{ij}$ is sampled uniformly from $\pm\overline{W}_{ij}$.

**Variational Bayesian neural nets.** Rather than fitting a point estimate of a neural net's weights, one can adopt the Bayesian approach of putting a prior distribution $p(W)$ over the weights and approximating the posterior distribution $p(W|\mathcal{D}) \propto p(W)p(\mathcal{D}|W)$, where $\mathcal{D}$ denotes the observed data. Graves (2011) observed that one could fit an approximation $q_{\boldsymbol{\theta}}(W) \approx p(W|\mathcal{D})$ using variational inference; in particular, one could maximize the evidence lower bound (ELBO) with respect to $\boldsymbol{\theta}$:

$$\mathcal{F}(\boldsymbol{\theta}) = \mathbb{E}_{W\sim q_{\boldsymbol{\theta}}}[\log p(\mathcal{D}\,|\,W)] - \mathrm{D_{KL}}(q_{\boldsymbol{\theta}}\,\|\,p).$$

The negation of the second term can be viewed as the description length of the data, and the negation of the first term can be viewed as the description length of the weights (Hinton & Van Camp, 1993). Graves (2011) observed that if $q$ is chosen to be a factorial Gaussian, sampling from $\boldsymbol{\theta}$ can be thought of as Gaussian weight perturbation where the variance is adapted to maximize $\mathcal{F}$. Blundell et al.

(2015) later combined this insight with the reparameterization trick (Kingma & Welling, 2014) to derive unbiased stochastic estimates of the gradient of $\mathcal{F}$.

**Evolution strategies.** ES (Rechenberg & Eigen, 1973) is a family of black box optimization algorithms which use weight perturbations to search for model parameters. ES was recently proposed as an alternative reinforcement learning algorithm (Schmidhuber et al., 2007; Salimans et al., 2017). In each iteration, ES generates a collection of weight perturbations as candidates and evaluates each according to a fitness function $F$. The gradient of the parameters can be estimated from the fitness function evaluations. ES is highly parallelizable, because perturbations can be generated and evaluated independently by different workers. Suppose $M$ is the number of workers, $\overline{W}$ is the model parameter, $\sigma$ is the standard deviation of the perturbations, $\alpha$ is the learning rate, $F$ is the objective function, and $\Delta W_m$ is the Gaussian noise generated at worker $m$. The ES algorithm tries to maximize $\mathbb{E}_{\Delta W} \left[ F \left( \overline{W} + \sigma \Delta W \right) \right]$. The gradient of the objective function and the update rule can be given as:

$$\nabla_{\overline{W}} \mathop{\mathbb{E}}_{\Delta W} \left[ F(\overline{W} + \Delta W) \right] = \frac{1}{\sigma^2} \mathop{\mathbb{E}}_{\Delta W} \left[ \Delta W F(\overline{W} + \Delta W) \right], \quad \text{where } \Delta W \sim \mathcal{N}(0, \sigma I)$$

$$\implies \overline{W}_{t+1} = \overline{W}_t + \alpha \frac{1}{M\sigma^2} \sum_{m=1}^{M} F(\overline{W}_t + \Delta W_m) \Delta W_m \tag{1}$$

## 2.2 Local Reparameterization Trick

In some cases, it's possible to reformulate weight perturbations as activation perturbations, thereby allowing them to be efficiently computed fully independently for different examples in a mini-batch. In particular, Kingma et al. (2015) showed that for fully connected networks with no weight sharing, unbiased stochastic gradients could be computed without explicit weight perturbations using the local reparameterization trick (LRT). For example, suppose $X$ is the input mini-batch, $W$ is the weight matrix and $B = XW$ is the matrix of activations. The LRT samples the activations $B$ rather than the weights $W$. In the case of a Gaussian posterior, the LRT is given by:

$$q_\theta(W_{i,j}) = \mathcal{N}(\mu_{i,j}, \sigma_{i,j}^2) \quad \forall W_{i,j} \in W \implies q_\theta(b_{m,j}|X) = \mathcal{N}(\gamma_{m,j}, \delta_{m,j})$$

$$\gamma_{m,j} = \sum_{i=1} x_{m,i} \mu_{i,j}, \quad \text{and} \quad \delta_{m,j} = \sum_{i=1} x_{m,i}^2 \sigma_{i,j}^2, \tag{2}$$

where $b_{m,j}$ denotes the perturbed activations. While the exact LRT applies only to fully connected networks with no weight sharing, Kingma et al. (2015) also introduced variational dropout, a regularization method inspired by the LRT which performs well empirically even for architectures the LRT does not apply to.

## 2.3 Other related work

Control variates are another general class of strategies for variance reduction, both for black-box optimization (Williams, 1992; Ranganath et al., 2014; Mnih & Gregor, 2014) and for gradient-based optimization (Roeder et al., 2016; Miller et al., 2017; Louizos et al., 2017). Control variates are complementary to flipout, so one could potentially combine these techniques to achieve a larger variance reduction. We also note that the fastfood transform (Le et al., 2013) is based on similar mathematical techniques. However, whereas fastfood is used to approximately multiply by a large Gaussian matrix, flipout preserves the random matrix's distribution and instead decorrelates the gradients between different samples.

## 3 Methods

As described above, weight perturbation algorithms suffer from high variance of the gradient estimates because all training examples in a mini-batch share the same perturbation. More precisely, sharing the perturbation induces correlations between the gradients, implying that the variance can't be eliminated by averaging. In this section, we introduce flipout, an efficient way to perturb the weights quasi-independently within a mini-batch.

### 3.1 FLIPOUT

We make two assumptions about the weight distribution $q_{\boldsymbol{\theta}}$: (1) the perturbations of different weights are independent; and (2) the perturbation distribution is symmetric around zero. These are nontrivial constraints, but they encompass important use cases: independent Gaussian perturbations (e.g. as used in variational BNNs and ES) and DropConnect with drop probability 0.5. We observe that, under these assumptions, the perturbation distribution is invariant to elementwise multiplication by a random sign matrix (i.e. a matrix whose entries are $\pm 1$). In the following, we denote elementwise multiplication by $\circ$.

**Observation 1.** Let $q_{\boldsymbol{\theta}}$ be a perturbation distribution that satisfies the above assumptions, and let $\widehat{\Delta W} \sim q_{\boldsymbol{\theta}}$. Let $E$ be a random sign matrix that is independent of $\widehat{\Delta W}$. Then $\Delta W = \widehat{\Delta W} \circ E$ is identically distributed to $\widehat{\Delta W}$. Furthermore, the loss gradients computed using $\Delta W$ are identically distributed to those computed using $\widehat{\Delta W}$.

Flipout exploits this fact by using a base perturbation $\widehat{\Delta W}$ shared by all examples in the mini-batch, and multiplies it by a different rank-one sign matrix for each example:

$$\Delta W_n = \widehat{\Delta W} \circ r_n s_n^\top, \tag{3}$$

where the subscript denotes the index within the mini-batch, and $r_n$ and $s_n$ are random vectors whose entries are sampled uniformly from $\pm 1$. According to Observation 1, the marginal distribution over gradients computed for individual training examples will be identical to the distribution computed using shared weight perturbations. Consequently, flipout yields an unbiased estimator for the loss gradients. However, by decorrelating the gradients between different training examples, we can achieve much lower variance updates when averaging over a mini-batch.

**Vectorization.** The advantage of flipout over explicit perturbations is that computations on a mini-batch can be written in terms of matrix multiplications. This enables efficient implementations on GPUs and modern accelerators such as the Tensor Processing Unit (TPU) (Jouppi et al., 2017). Let $x$ denote the activations in one layer of a neural net. The next layer's activations are given by:

$$
\begin{aligned}
y_n &= \phi\left(W^\top x_n\right) \\
&= \phi\left(\left(\overline{W} + \widehat{\Delta W} \circ r_n s_n^\top\right)^\top x_n\right) \\
&= \phi\left(\overline{W}^\top x_n + \left(\widehat{\Delta W}^\top (x_n \circ s_n)\right) \circ r_n\right),
\end{aligned}
$$

where $\phi$ denotes the activation function. To vectorize these computations, we define matrices $R$ and $S$ whose rows correspond to the random sign vectors $r_n$ and $s_n$ for all examples in the mini-batch. The above equation is vectorized as:

$$Y = \phi\left(X\overline{W} + \left((X \circ S)\widehat{\Delta W}\right) \circ R\right). \tag{4}$$

This defines the forward pass. Because $R$ and $S$ are sampled independently of $\overline{W}$ and $\widehat{\Delta W}$, we can backpropagate through Eqn. 4 to obtain derivatives with respect to $\overline{W}$, $\widehat{\Delta W}$, and $X$.

**Computational cost.** In general, the most expensive operation in the forward pass is matrix multiplication. Flipout's forward pass requires two matrix multiplications instead of one, and therefore should be roughly twice as expensive as a forward pass with a single shared perturbation when the multiplications are done in sequence.[1] However, note that the two matrix multiplications are independent and can be done in parallel; this incurs the same overhead as the local reparameterization trick (Kingma et al., 2015).

A general rule of thumb for neural nets is that the backward pass requires roughly twice as many FLOPs as the forward pass. This suggests that each update using flipout ought to be about twice as expensive as an update with a single shared perturbation (if the matrix multiplications are done sequentially); this is consistent with our experience.

---

[1]Depending on the efficiency of the underlying libraries, the overhead of sampling $R$ and $S$ may be non-negligible. If this is an issue, these matrices may be reused between all mini-batches. In our experience, this does not cause any drop in performance.

**Evolution strategies.** ES is a highly parallelizable algorithm; however, most ES systems are engineered to run on multi-core CPU machines and are not able to take full advantage of GPU parallelism. Flipout enables ES to run more efficiently on a GPU because it allows each worker to evaluate a batch of quasi-independent perturbations rather than only a single perturbation. To apply flipout to ES, we can simply replicate the starting state by the number of flipout perturbations $N$, at each worker. Instead of Eqn. 1, the update rule using $M$ workers becomes:

$$\overline{W}_{t+1} = \overline{W}_t + \alpha \frac{1}{MN\sigma^2} \sum_{m=1}^{M} \sum_{n=1}^{N} F_{mn} \left\{ \widehat{\Delta W}_m \circ r_{mn} s_{mn}^\top \right\} \tag{5}$$

where $m$ indexes workers, $n$ indexes the examples in a worker's batch, and $F_{mn}$ is the reward evaluated with the $n^{th}$ perturbation at worker $m$. Hence, each worker is able to evaluate multiple perturbations as a batch, allowing for parallelism on a GPU architecture.

## 3.2 Variance analysis

In this section, we analyze the variance of stochastic gradients with and without flipout. We show that flipout is guaranteed to reduce the variance of the gradient estimates compared to using naïve shared perturbations.

Let $\mathcal{G}_x = \mathcal{G}(x, \Delta W) = \frac{\partial}{\partial \theta_i} \mathcal{L}(y, f(x, \overline{W}, \Delta W))$ denote one entry of the stochastic gradient $\nabla_{\boldsymbol{\theta}} \mathcal{L}(y, f(x, \overline{W}, \Delta W))$ under the perturbation $\Delta W$ for a single training example $x$. (Note that $\mathcal{G}_x$ is a random variable which depends on both $x$ and $\Delta W$. We analyze a single entry of the gradient so that we can work with scalar-valued variances.) We denote the gradient averaged over a mini-batch as the random variable $\mathcal{G}_\mathcal{B} = \frac{1}{N} \sum_{n=1}^{N} \mathcal{G}(x_n, \Delta W_n)$, where $\mathcal{B} = \{x_n\}_{n=1}^{N}$ denotes a mini-batch of size $N$, and $\Delta W_n$ denotes the perturbation for the $n^{th}$ example. (The randomness comes from both the choice of $\mathcal{B}$ and the random perturbations.) For simplicity, we assume that the $x_n$ are sampled i.i.d. from the data distribution.

Using the Law of Total Variance, we decompose $\mathrm{Var}(\mathcal{G}_\mathcal{B})$ into a data term (the variance of the exact mini-batch gradients) and an estimation term (the estimation variance for a fixed mini-batch):

$$\mathrm{Var}\left(\mathcal{G}_\mathcal{B}\right) = \underbrace{\mathrm{Var}_\mathcal{B}\left(\mathop{\mathbb{E}}_{\Delta W}\left[\mathcal{G}_\mathcal{B} \,|\, \mathcal{B}\right]\right)}_{\text{data}} + \underbrace{\mathop{\mathbb{E}}_\mathcal{B}\left[\mathrm{Var}_{\Delta W}\left(\mathcal{G}_\mathcal{B} \,|\, \mathcal{B}\right)\right]}_{\text{estimation}}. \tag{6}$$

Notice that the data term decays with $N$ while the estimation term may not, due to its dependence on the shared perturbation. But we can break the estimation term into two parts for which we can analyze the dependence on $N$. To do this, we reformulate the standard shared perturbation scheme as follows: $\Delta W$ is generated by first sampling $\widehat{\Delta W}$ and then multiplying it by a random sign matrix $rs^\top$ as in Eqn. 3 — exactly like flipout, except that the sign matrix is shared by the whole mini-batch. According to Observation 1, this yields an identical distribution for $\Delta W$ to the standard shared perturbation scheme. Based on this, we obtain the following decomposition:

**Theorem 2** (Variance Decomposition Theorem). *Define $\alpha$, $\beta$, and $\gamma$ to be*

$$\alpha = \mathrm{Var}_x\left(\mathop{\mathbb{E}}_{\Delta W}\left[\mathcal{G}_x \,|\, x\right]\right) + \mathop{\mathbb{E}}_x\left[\mathrm{Var}_{\Delta W}(\mathcal{G}_x \,|\, x)\right] \tag{7}$$

$$\beta = \mathop{\mathbb{E}}_{x,x',\widehat{\Delta W}}\left[\mathrm{Cov}_{\Delta W}(\mathcal{G}_x, \mathcal{G}_{x'} \,|\, x, x', \widehat{\Delta W})\right] \tag{8}$$

$$\gamma = \mathop{\mathbb{E}}_{x,x'}\left[\mathrm{Cov}_{\widehat{\Delta W}}\left(\mathop{\mathbb{E}}_{\Delta W}[\mathcal{G}_x \,|\, x, \widehat{\Delta W}], \mathop{\mathbb{E}}_{\Delta W}[\mathcal{G}_{x'} \,|\, x', \widehat{\Delta W}] \,|\, x, x'\right)\right] \tag{9}$$

*Under the assumptions of Observation 1, the variance of the gradients under shared perturbations and flipout perturbations can be written in terms of $\alpha$, $\beta$, and $\gamma$ as follows:*

$$\textit{Fully independent perturbations: } \mathrm{Var}(\mathcal{G}_\mathcal{B}) = \frac{1}{N}\alpha \tag{10}$$

$$\textit{Shared perturbation: } \mathrm{Var}(\mathcal{G}_\mathcal{B}) = \frac{1}{N}\alpha + \frac{N-1}{N}(\beta + \gamma) \tag{11}$$

$$\textit{Flipout: } \mathrm{Var}(\mathcal{G}_\mathcal{B}) = \frac{1}{N}\alpha + \frac{N-1}{N}\gamma \tag{12}$$

*Proof.* Details of the proof are provided in Appendix A. □

We can interpret $\alpha$, $\beta$, and $\gamma$ as follows. First, $\alpha$ combines the data term from Eqn. 6 with the expected estimation variance for individual data points. This corresponds to the variance of the gradients on individual training examples, so fully independent perturbations yield a total variance of $\alpha/N$. The other terms, $\beta$ and $\gamma$, reflect the covariance between the estimation errors on different training examples as a result of the shared perturbations. The term $\beta$ reflects the covariance that results from sampling $r$ and $s$, so it is eliminated by flipout, which samples these vectors independently. Finally, $\gamma$ reflects the covariance that results from sampling $\widehat{\Delta W}$, which flipout does not eliminate.

Empirically, for all the neural networks we investigated, we found that $\alpha \gg \beta \gg \gamma$. This implies the following behavior for $\text{Var}(\mathcal{G}_\mathcal{B})$ as a function of $N$: for small $N$, the data term $\alpha/N$ dominates, giving a $1/N$ variance reduction; with shared perturbations, once $N$ is large enough that $\alpha/N < \beta$, the variance $\text{Var}(\mathcal{G}_\mathcal{B})$ levels off to $\beta$. However, flipout continues to enjoy a $1/N$ variance reduction in this regime. In principle, flipout's variance should level off at the point where $\alpha/N < \gamma$, but in all of our experiments, $\gamma$ was small enough that this never occurred: flipout's variance was approximately $\alpha/N$ throughout the entire range of $N$ values we explored, just as if the perturbations were sampled fully independently for every training example.

## 4 EXPERIMENTS

We first verified empirically the variance reduction effect of flipout predicted by Theorem 2; we measured the variance of the gradients under different perturbations for a wide variety of neural network architectures and batch sizes. In Section 4.2, we show that flipout applied to Gaussian perturbations and DropConnect is effective at regularizing LSTM networks. In Section 4.3, we demonstrate that flipout converges faster than shared perturbations when training with large mini-batches. Finally, in Section 4.4 we present experiments combining Evolution Strategies with flipout in both supervised learning and reinforcement learning tasks.

In our experiments, we consider the four architectures shown in Table 1 (details in Appendix B).

### 4.1 VARIANCE REDUCTION

Since the main effect of flipout is intended to be variance reduction of the gradients, we first estimated the gradient variances of several architectures with mini-batch sizes ranging from 1 to 8196 (Fig. 1). We experimented with three perturbation methods: a single shared perturbation per mini-batch, the local reparameterization trick (LRT) of Kingma et al. (2015), and flipout.

For each of the FC, ConVGG, and LSTM architectures, we froze a partially trained network to use for all variance estimates, and we used multiplicative Gaussian perturbations with $\sigma^2 = 1$. We computed Monte Carlo estimates of the gradient variance, including both the data and estimation terms in Eqn. 6. Confidence intervals are based on 50 independent runs of the estimator. Details are given in Appendix C.

The analysis in Section 3.2 makes strong predictions about the shapes of the curves in Fig. 1. By Theorem 2, the variance curves for flipout and shared perturbations each have the form $a + b/N$, where $N$ is the mini-batch size. On a log-log plot, this functional form appears as a linear regime with slope -1, a constant regime, and a smooth phase transition in between. Also, because the distribution of individual gradients is identical with and without flipout, the curves must agree for $N = 1$.

| Name | Network Type | Data Set |
|---|---|---|
| ConvLe | (Shallow) Convolutional | MNIST (LeCun et al., 1998) |
| ConVGG | (Deep) Convolutional | CIFAR-10 (Krizhevsky & Hinton, 2009) |
| FC | Fully Connected | MNIST |
| LSTM | LSTM Network | Penn Treebank (Marcus et al., 1993) |

Table 1: Network Configurations

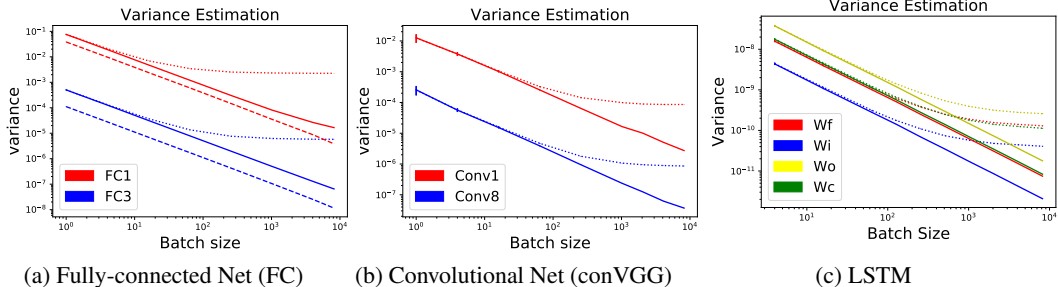

Figure 1: Empirical variance of gradients with respect to mini-batch size for several architectures. **(a)** FC on MNIST; FC1 denotes the first layer of the FC network. **(b)** ConVGG on CIFAR-10; Conv1 denotes the first convolutional layer. **(c)** LSTM on Penn Treebank; the variance is shown for the hidden-to-hidden weight matrices in the first LSTM layer: $W_f$, $W_i$, $W_o$, and $W_c$ are the weights for the *forget, input* and *output* gates, and the *candidate cell* update, respectively. **Dotted:** shared perturbations. **Solid:** flipout. **Dashed:** LRT.

Our plots are consistent with both of these predictions. We observe that for shared perturbations, the phase transition consistently occurs for mini-batch sizes somewhere between 100 and 1000. In contrast, flipout gives the ideal linear variance reduction throughout the range of mini-batch sizes we investigated, i.e., its behavior is indistinguishable from fully independent perturbations.

As analyzed by Kingma et al. (2015), the LRT gradients are fully independent within a mini-batch, and are therefore guaranteed to achieve the ideal $1/N$ variance reduction. Furthermore, they reduce the variance below that of explicit weight perturbations, so we would expect them to achieve smaller variance than flipout, as shown in Fig. 1a. However, flipout is applicable to a wider variety of architectures, including convolutional nets and RNNs.

## 4.2 REGULARIZATION FOR LANGUAGE MODELING

We evaluated the regularization effect of flipout on the character-level and word-level language modeling tasks with the Penn Treebank corpus (PTB) (Marcus et al., 1993). We compared flipout to several other methods for regularizing RNNs: naïve dropout (Zaremba et al., 2014), variational dropout (Gal & Ghahramani, 2016), recurrent dropout (Semeniuta et al., 2016), zoneout (Krueger et al., 2016), and DropConnect (Merity et al., 2017). Zaremba et al. (2014) apply dropout only to the feed-forward connections of an RNN (to the input, output, and connections between layers). The other methods regularize the recurrent connections as well: Semeniuta et al. (2016) apply dropout to the cell update vector, with masks sampled either per step or per sequence; Gal & Ghahramani (2016) apply dropout to the forward and recurrent connections, with all dropout masks sampled per sequence. Merity et al. (2017) use DropConnect to regularize the hidden-to-hidden weight matrices, with a single DropConnect mask shared between examples in a mini-batch. We denote their model WD (for weight-dropped LSTM).

**Character-Level.** For our character-level experiments, we used a single-layer LSTM with 1000 hidden units. We trained each model on non-overlapping sequences of 100 characters in batches of size 32, using the AMSGrad variant of Adam (Reddi et al., 2018) with learning rate 0.002. We perform early stopping based on validation performance. Here, we applied flipout to the hidden-to-hidden weight matrix. More hyperparameter details are given in Appendix D. The results, measured in bits-per-character (BPC) for the validation and test sequences of PTB, are shown in Table 2. In the table, shared perturbations and flipout (with Gaussian noise sampling) are denoted by Mult. Gauss and Mult. Gauss + Flipout, respectively. We also compare to RBN (recurrent batchnorm) (Cooijmans et al., 2017) and H-LSTM+LN (HyperLSTM + LayerNorm) (Ha et al., 2016). Mult. Gauss + Flipout outperforms the other methods, and achieves the best reported results for this architecture.

**Word-Level.** For our word-level experiments, we used a 2-layer LSTM with 650 hidden units per layer and 650-dimensional word embeddings. We trained on sequences of length 35 in batches of size 40, for 100 epochs. We used SGD with initial learning rate 30, and decayed the learning rate by a factor of 4 based on the nonmonotonic criterion introduced by Merity et al. (2017). We used flipout to implement DropConnect, as described in Section 2.1, and call this WD+Flipout. We applied WD+Flipout to the hidden-to-hidden weight matrices for recurrent regularization, and used the same hyperparameters as Merity et al. (2017). We used embedding dropout (setting rows of the embedding matrix to 0) with probability 0.1 for all regularized models except Gal, where we used

| Model | Valid | Test |
|---|---|---|
| Unregularized LSTM | 1.468 | 1.423 |
| Semeniuta (2016) | 1.337 | 1.300 |
| Zoneout (2016) | 1.306 | 1.270 |
| Gal (2016) | 1.277 | 1.245 |
| Mult. Gauss ($\sigma = 1$) (ours) | 1.257 | 1.230 |
| Mult. Gauss + Flipout (ours) | **1.256** | **1.227** |
| RBN (2017) | – | 1.32 |
| H-LSTM + LN (2016) | 1.281 | 1.250 |

Table 2: Bits-per-character (BPC) for the character-level PTB task. The RBN and H-LSTM+LN results are from the respective papers. All other results are from our own experiments.

| Model | Valid | Test |
|---|---|---|
| Unregularized LSTM | 132.23 | 128.97 |
| Zaremba (2014) | 80.40 | 76.81 |
| Semeniuta (2016) | 81.91 | 77.88 |
| Gal (2016) | 78.24 | 75.39 |
| Zoneout (2016) | 78.66 | 75.45 |
| WD (2017) | 78.82 | 75.71 |
| WD + Flipout (ours) | **76.88** | **73.20** |

Table 3: Perplexity on the PTB word-level validation and test sets. All results are from our own experiments.

probability 0.2 as specified in their paper. More hyperparameter details are given in Appendix D. We show in Table 3 that WD+Flipout outperforms the other methods with respect to both validation and test perplexity. In Appendix E.4, we show that WD+Flipout yields significant variance reduction for large mini-batches, and that when training with batches of size 8192, it converges faster than WD.

### 4.3 LARGE BATCH TRAINING WITH FLIPOUT

Theorem 2 and Fig. 1 suggest that the variance reduction effect of flipout is more pronounced in the large mini-batch regime. In this section, we train a Bayesian neural network with mini-batches of size 8192 and show that flipout speeds up training in terms of the number of iterations.

We trained the FC and ConvLe networks from Section 4.1 using Bayes by Backprop (Blundell et al., 2015). Since our primary focus is optimization, we focus on the training loss, shown in Fig. 2a: for FC, we compare flipout with shared perturbations and the LRT; for ConvLe, we compare only to shared perturbations since the LRT does not give an unbiased gradient estimator. We found that flipout converged in about 3 times fewer iterations than shared perturbations for both models, while achieving comparable performance to the LRT for the FC model. Because flipout is roughly twice as expensive as shared perturbations (see Section 3.1), this corresponds to a 1.5x speedup overall. Curves for the training and test error are given in Appendix E.2.

### 4.4 EVOLUTION STRATEGIES

ES typically runs on multiple CPU cores. The challenge in making ES GPU-friendly is that each sample requires computing a separate weight perturbation, so traditionally each worker can only generate one sample at a time. In Section 3.1, we showed that ES with flipout allows each worker to evaluate a batch of perturbations, which can be done efficiently on a GPU. However, flipout induces correlations between the samples, so we investigated whether these correlations cause a slowdown in training relative to fully independent perturbations (which we term "IdealES"). In this section, we show empirically that flipout ES is just as sample-efficient as IdealES, and consequently one can obtain significantly higher throughput per unit cost using flipout ES on a GPU.

The ES gradient defined in Eqn. 1 has high variance, so a large number of samples are generally needed before applying an update. We found that 5,000 samples are needed to achieve stable performance in the supervised learning tasks. Standard ES runs the forward pass 5,000 times with independent weight perturbations, which sees little benefit to using a GPU over a CPU. FlipES allows the same number of samples to be evaluated using a much smaller number of explicit perturbations. Throughout the experiments, we ran flipout with mini-batches of size 40 (i.e. $N = 40$ in Eqn. 5).

We compared IdealES and FlipES with a fully connected network (FC) on the MNIST dataset. Fig. 2b shows that we incur no loss in performance when using pseudo-independent noise. Next, we compared FlipES and cpuES (using 40 CPU cores) in terms of the per-update time with respect to the model size. The result (in Appendix E.3) shows that FlipES scales better because it runs on the GPU. Finally, we compared FlipES and the backpropagation algorithm on both FC and ConvLe. Fig. 2c and Fig. 2d show that FlipES achieves data efficiency comparable with the backpropagation algorithm. IdealES has a much higher computational cost than backpropagation, due to the large number of forward passes. FlipES narrows the computational gap between them. Although ES is

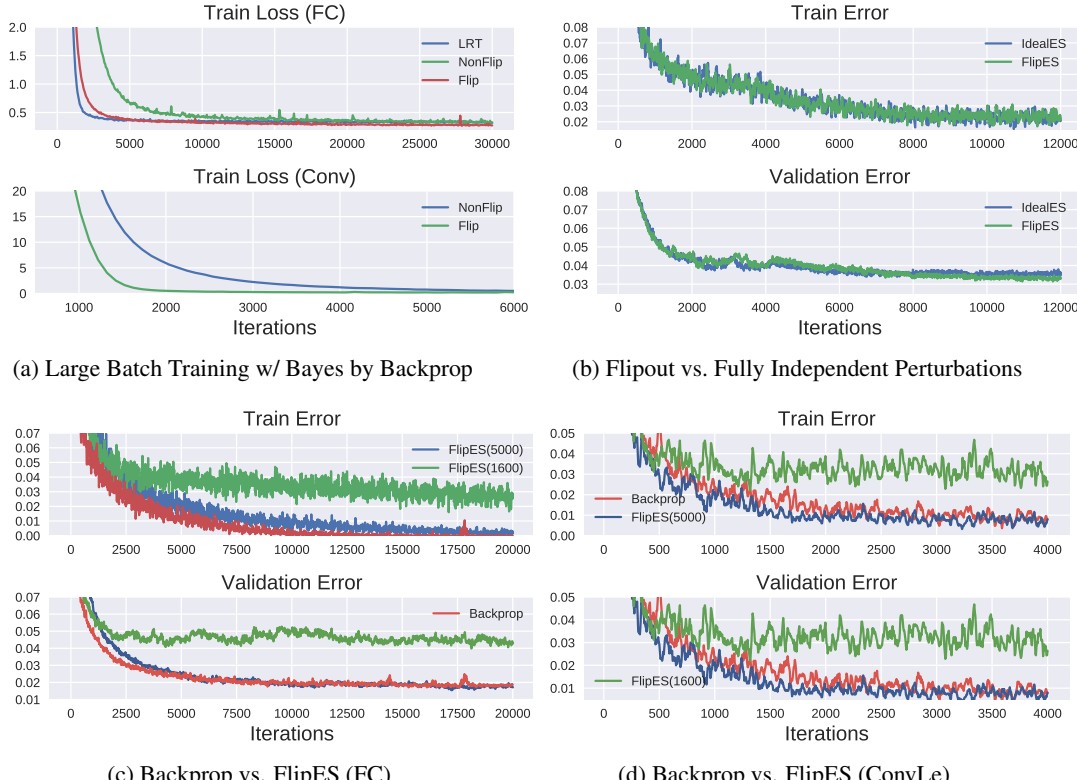

Figure 2: Large batch training and ES. **a)** Training loss per iteration using Bayes By Backprop with batch size 8192 on the FC and ConvLe networks. **b)** Error rate of the FC network on MNIST using ES with 1,600 samples per update; there is no drop in performance compared to ideal ES. **c)** Error rate of FC on MNIST, comparing FlipES (with either 5,000 or 1,600 samples per update) with backpropagation. (This figure does not imply that FlipES is more efficient than backprop; FlipES was around 60 times more expensive than backprop per update.) **d)** The same as (c), except run on ConvLe.

more expensive than backpropagation, it can be applied to models which are not fully differentiable, such as models with a discrete loss (e.g., accuracy or BLEU score) or with stochastic units.

## 5  CONCLUSIONS

We have introduced flipout, an efficient method for decorrelating the weight gradients between different examples in a mini-batch. We showed that flipout is guaranteed to reduce the variance compared with shared perturbations. Empirically, we demonstrated significant variance reduction in the large batch setting for a variety of network architectures, as well as significant speedups in training time. We showed that flipout outperforms dropout-based methods for regularizing LSTMs. Flipout also makes it practical to apply GPUs to evolution strategies, resulting in substantially increased throughput for a given computational cost. We believe flipout will make weight perturbations practical in the large batch setting favored by modern accelerators such as Tensor Processing Units (Jouppi et al., 2017).

## ACKNOWLEDGMENTS

YW was supported by an NSERC USRA award, and PV was supported by a Connaught New Researcher Award. We thank David Duvenaud, Alex Graves, Geoffrey Hinton, and Matthew D. Hoffman for helpful discussions.

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

# A PROOF OF THEOREM 2

In this section, we provide the proof of Theorem 2 (Variance Decomposition Theorem).

*Proof.* We use the notations from Section 3.2. Let $x, x'$ denote two training examples from the mini-batch $\mathcal{B}$, and $\Delta W, \Delta W'$ denote the weight perturbations they received. We begin with the decomposition into data and estimation terms (Eqn. 6), which we repeat here for convenience:

$$\mathrm{Var}\left(\mathcal{G}_{\mathcal{B}}\right) = \underbrace{\mathrm{Var}_{\mathcal{B}}\left(\mathbb{E}_{\Delta W}\left[\mathcal{G}_{\mathcal{B}} \mid \mathcal{B}\right]\right)}_{\text{data}} + \underbrace{\mathbb{E}_{\mathcal{B}}\left[\mathrm{Var}_{\Delta W}\left(\mathcal{G}_{\mathcal{B}} \mid \mathcal{B}\right)\right]}_{\text{estimation}}. \tag{13}$$

The data term from Eqn. 13 can be simplified:

$$\mathrm{Var}_{\mathcal{B}}\left(\mathbb{E}_{\Delta W}\left[\mathcal{G}_{\mathcal{B}} \mid \mathcal{B}\right]\right) = \mathrm{Var}_{\mathcal{B}}\left(\mathbb{E}_{\Delta W}\left[\frac{1}{N}\sum_{n=1}^{N}\mathcal{G}_{x_n} \mid \mathcal{B}\right]\right)$$

$$= \mathrm{Var}_{\mathcal{B}}\left(\frac{1}{N}\sum_{n=1}^{N}\mathbb{E}_{\Delta W}\left[\mathcal{G}_{x_n} \mid x_n\right]\right)$$

$$= \frac{1}{N}\mathrm{Var}_{x}\left(\mathbb{E}_{\Delta W}\left[\mathcal{G}_{x} \mid x\right]\right) \tag{14}$$

We break the estimation term from Eqn. 13 into variance and covariance terms:

$$\mathbb{E}_{\mathcal{B}}\left[\mathrm{Var}_{\Delta W}\left(\mathcal{G}_{\mathcal{B}} \mid \mathcal{B}\right)\right] = \mathbb{E}_{\mathcal{B}}\left[\mathrm{Var}_{\Delta W}\left(\frac{1}{N}\sum_{n=1}^{N}\mathcal{G}_{x_n} \mid x_n\right)\right]$$

$$= \frac{1}{N^2}\mathbb{E}_{\mathcal{B}}\left[\sum_{n=1}^{N}\sum_{n'=1}^{N}\mathrm{Cov}_{\Delta W_n,\Delta W_{n'}}\left(\mathcal{G}_{x_n},\mathcal{G}_{x_{n'}} \mid x_n, x_{n'}\right)\right]$$

$$= \frac{1}{N^2}\mathbb{E}_{\mathcal{B}}\left[\sum_{n=1}^{N}\mathrm{Var}_{\Delta W_n}\left(\mathcal{G}_{x_n} \mid x_n\right) + \sum_{n \neq n'}\mathrm{Cov}_{\Delta W_n,\Delta W_{n'}}\left(\mathcal{G}_{x_n},\mathcal{G}_{x_{n'}} \mid x_n, x_{n'}\right)\right]$$

$$= \frac{1}{N}\mathbb{E}_{x}\left[\mathrm{Var}_{\Delta W}(\mathcal{G}_{x} \mid x)\right] + \frac{N-1}{N}\mathbb{E}_{x,x'}\left[\mathrm{Cov}_{\Delta W,\Delta W'}(\mathcal{G}_{x},\mathcal{G}_{x'} \mid x, x')\right] \tag{15}$$

We now separately analyze the cases of fully independent perturbations, shared perturbations, and flipout.

**Fully independent perturbations.** If the perturbations are fully independent, the second term in Eqn. 15 disappears. Hence, combining Eqns. 13, 14, and 15, we are left with

$$\mathrm{Var}(\mathcal{G}_{\mathcal{B}}) = \frac{1}{N}\mathrm{Var}_{x}\left(\mathbb{E}_{\Delta W}\left[\mathcal{G}_{x} \mid x\right]\right) + \frac{1}{N}\mathbb{E}_{x}\left[\mathrm{Var}_{\Delta W}(\mathcal{G}_{x} \mid x)\right], \tag{16}$$

which is just $\alpha/N$.

**Shared perturbations.** Recall that we reformulate the shared perturbations in terms of first sampling $\widehat{\Delta W}$, and then letting $\Delta W = \widehat{\Delta W} \circ rs^\top$, where $r$ and $s$ are random sign vectors shared by the whole batch. Using the Law of Total Variance, we break the second term in Eqn. 15 into a part that comes from sampling $\widehat{\Delta W}$ and a part that comes from sampling $r$ and $s$.

$$\mathrm{Cov}_{\Delta W,\Delta W'}(\mathcal{G}_{x},\mathcal{G}_{x'} \mid x, x') = \mathbb{E}_{\widehat{\Delta W}}\left[\mathrm{Cov}_{\Delta W,\Delta W'}(\mathcal{G}_{x},\mathcal{G}_{x'} \mid x, x', \widehat{\Delta W}) \mid x, x'\right] +$$

$$+ \mathrm{Cov}_{\widehat{\Delta W}}\left(\mathbb{E}_{\Delta W}[\mathcal{G}_{x} \mid x, \widehat{\Delta W}], \mathbb{E}_{\Delta W'}[\mathcal{G}_{x'} \mid x', \widehat{\Delta W}] \mid x, x'\right) \tag{17}$$

Since the perturbations are shared, $\Delta W' = \Delta W$, so this can be simplified slightly to:

$$\underset{\widehat{\Delta W}}{\mathbb{E}} \left[ \underset{\Delta W}{\text{Cov}}(\mathcal{G}_x, \mathcal{G}_{x'} \,|\, x, x', \widehat{\Delta W}) \,|\, x, x' \right] + \underset{\widehat{\Delta W}}{\text{Cov}} \left( \underset{\Delta W}{\mathbb{E}}[\mathcal{G}_x \,|\, x, \widehat{\Delta W}], \underset{\Delta W}{\mathbb{E}}[\mathcal{G}_{x'} \,|\, x', \widehat{\Delta W}] \,|\, x, x' \right) \quad (18)$$

Plugging these two terms into the second term of Eqn. 15 yields $\frac{N-1}{N}(\beta + \gamma)$, so putting this all together we get $\text{Var}(\mathcal{G}_\mathcal{B}) = \frac{1}{N}\alpha + \frac{N-1}{N}(\beta + \gamma)$.

**Flipout.** Since the perturbations for different examples are independent conditioned on $\widehat{\Delta W}$, the first term of Eqn. 17 vanishes. However, the second term remains. Therefore, plugging this into Eqn. 15 and combining the result with Eqns. 13 and 14, we are left with $\text{Var}(\mathcal{G}_\mathcal{B}) = \frac{1}{N}\alpha + \frac{N-1}{N}\gamma$.

□

## B  NETWORK CONFIGURATIONS

Here, we provide details of the network configurations used for our experiments (Section 4).

The FC network is a 3-layer fully-connected network with 512-512-10 hidden units.

ConvLe is a LeNet-like network (LeCun et al., 1998) where the first two layers are convolutional with 32 and 64 filters of size $[5, 5]$, and use ReLU non-linearities. A $2 \times 2$ max pooling layer follows after each convolutional layer. Dimensionality reduction only takes place in the pooling layer; the stride for pooling is two and padding is used in the convolutional layers to keep the dimension. Two fully-connected layers with 1024 and 10 hidden units are used to produce the classification result.

ConVGG is based on the VGG16 network (Simonyan & Zisserman, 2014). We modified the last fully connected layer to have 10 output dimensions for our experiments on CIFAR-10. We didn't use batch normalization for the variance reduction experiment since it introduces extra stochasticity.

The architectures used for the LSTM experiments are described in Section 4.2. The hyperparameters used for the language modelling experiments are provided in Appendix D.

## C  VARIANCE REDUCTION EXPERIMENT DETAILS

Given a network architecture, we compute the empirical stochastic gradient update variance as follows. We start with a moderately pre-trained model, such as a network with $85\%$ training accuracy on MNIST. Without updating the parameters, we obtain the gradients of all the weights by performing a feed-forward pass, that includes sampling $\widehat{\Delta W}$, $R$, and $S$, followed by backpropagation. The gradient variance of each weight is computed by repeating this procedure 200 times in the experiments. Let $\widetilde{\text{Var}}_{lj}$ denote the estimate of the gradient variance of weight $j$ in layer $l$. We compute the gradient variance as follows:

$$\widetilde{\text{Var}}_{lj} = \frac{1}{200} \sum_{i=1}^{200} (g_{lj}^i - \overline{g_{lj}})^2 \quad \text{where} \quad \overline{g_{lj}} = \frac{1}{200} \sum_{i=1}^{200} g_{lj}^i$$

where $g_{lj}^i$ is the gradient received by weight $j$ in layer $l$. We estimate the variance of the gradients in layer $l$ by averaging the variances of the weights in that layer, $\tilde{V} = \frac{1}{|J|} \sum_j \widetilde{\text{Var}}_{lj}$. In order to compute a confidence interval on the gradient variance estimate, we repeat the above procedure 50 times, yielding a sequence of average variance estimates, $\widetilde{V_1}, \dots, \widetilde{V_{50}}$. For Fig. 1, we compute the $90\%$ confidence intervals of the variance estimates with a t-test.

For ConVGG, multiple GPUs were needed to run the variance reduction experiment with large mini-batch sizes (such as 4096 and 8192). In such cases, it is computationally efficient to generate independent weight perturbations on different GPUs. However, since our aim was to understand the effects of variance reduction independent of implementation, we shared the base perturbation among all GPUs to produce the plot shown in Fig. 1. We show in Appendix E that flipout yields lower variance even when we sample independent perturbations on different GPUs.

For the LSTM variance reduction experiments, we used the two-layer LSTM described in Section 4.2, trained for 3 epochs on the word-level Penn Treebank dataset. For Fig. 1, we split large

mini-batches (size 128 and higher) into sub-batches of size 64; we sampled one base perturbation $\Delta W$ that was shared among all sub-batches, and we sampled independent $R$ and $S$ matrices for each sub-batch.

## D  LSTM Regularization Experiment Details

Long Short-Term Memory networks (LSTMs) are defined by the following equations:

$$\mathbf{i}_t, \mathbf{f}_t, \mathbf{o}_t = \sigma(\mathbf{W}_h \mathbf{h}_{t-1} + \mathbf{W}_x \mathbf{x}_t + \mathbf{b}) \tag{19}$$

$$\mathbf{g}_t = \tanh(\mathbf{W}_g \mathbf{h}_{t-1} + \mathbf{U}_g \mathbf{x}_t + \mathbf{b}_g) \tag{20}$$

$$\mathbf{c}_t = \mathbf{f}_t \circ \mathbf{c}_{t-1} + \mathbf{i}_t \circ \mathbf{g}_t \tag{21}$$

$$\mathbf{h}_t = \mathbf{o}_t \circ \tanh(\mathbf{c}_t) \tag{22}$$

where $\mathbf{i}_t$, $\mathbf{f}_t$, and $\mathbf{o}_t$ are the input, forget, and output gates, respectively, $\mathbf{g}_t$ is the candidate update, and $\circ$ denotes elementwise multiplication. Naïve application of dropout on the hidden state of an LSTM is not effective, because it leads to significant memory loss over long sequences. Several approaches have been proposed to regularize the recurrent connections, based on applying dropout to specific terms in the LSTM equations. Semeniuta et al. (2016) propose to drop the cell update vector, with a dropout mask $\mathbf{d_t}$ sampled either per-step or per-sequence: $\mathbf{c}_t = \mathbf{f}_t \circ \mathbf{c}_{t-1} + \mathbf{i}_t \circ (\mathbf{d_t} \circ \mathbf{g}_t)$. Gal & Ghahramani (2016) apply dropout to the input and hidden state at each time step, $\mathbf{x}_t \circ \mathbf{d_x}$ and $\mathbf{h_{t-1}} \circ \mathbf{d_h}$, with dropout masks $\mathbf{d_x}$ and $\mathbf{d_h}$ sampled once per sequence (and repeated in each time step). Krueger et al. (2016) propose to *zone out* units rather than dropping them; the hidden state and cell values are either stochastically updated or maintain their previous value: $\mathbf{c}_t = \mathbf{d_t^c} \circ \mathbf{c_{t-1}} + (1 - \mathbf{d_t^c}) \circ (\mathbf{f_t} \circ \mathbf{c_{t-1}} + \mathbf{i_t} \circ \mathbf{g_t})$ and $\mathbf{h}_t = \mathbf{d_t^h} \circ \mathbf{h_{t-1}} + (1 - \mathbf{d_t^h}) \circ (\mathbf{o_t} \circ \tanh(\mathbf{f_t} \circ \mathbf{c_{t-1}} + \mathbf{i_t} \circ \mathbf{g_t}))$, with zoneout masks $\mathbf{d_t^h}$ and $\mathbf{d_t^c}$ sampled per step.

### D.1  Hyperparameter Details

For the word-level models (Table 3), we used gradient clipping threshold 0.25 and the following hyperparameters:

- For Gal & Ghahramani (2016), we used variational dropout with the parameters given in their paper: 0.35 dropout probability on inputs and outputs, 0.2 hidden state dropout, and 0.2 embedding dropout.

- For Semeniuta et al. (2016), we used 0.1 embedding dropout, 0.5 dropout on inputs and outputs, and 0.3 dropout on cell updates, with per-step mask sampling.

- For Krueger et al. (2016), we used 0.1 embedding dropout, 0.5 dropout on inputs and outputs, and cell and hidden state zoneout probabilities of 0.25 and 0.025, respectively.

- For WD (Merity et al., 2017), we used the parameters given in their paper: 0.1 embedding dropout, 0.4 dropout probability on inputs and outputs, and 0.3 dropout probability on the output between layers (the same masks are used for each step of a sequence). We use 0.5 probability for DropConnect applied to the hidden-to-hidden weight matrices.

- For WD+Flipout, we used the same parameters as Merity et al. (2017), given above, but we regularized the hidden-to-hidden weight matrices with the variant of flipout described in Section 2.1, which implements DropConnect with probability 0.5.

For the character-level models (Table 2), we used orthogonal initialization for the LSTM weight matrices, gradient clipping threshold 1, and did not use input or output dropout. The input characters were represented as one-hot vectors. We used the following hyperparameters for each model:

- For recurrent dropout (Semeniuta et al., 2016), we used 0.25 dropout probability on the cell state, and per-step mask sampling.

- For Zoneout (Krueger et al., 2016), we used 0.5 and 0.05 for the cell and hidden state zoneout probabilities, respectively.

- For the variational LSTM (Gal & Ghahramani, 2016), we used 0.25 hidden state dropout.

- For the flipout and shared perturbation LSTMs, we sampled Gaussian noise with $\sigma = 1$ for the hidden-to-hidden weight matrix.

# E  ADDITIONAL EXPERIMENTS

## E.1  VARIANCE REDUCTION

As discussed in Appendix B, training on multiple GPUs naturally induces independent noise for each sub-batch. Fig. 3 shows that flipout still achieves lower variance than shared perturbations in such cases. When estimating the variance with mini-batch size 8192, running on four GPUs naturally induces four independent noise samples, for each sub-batch of size 2048; this yields lower variance than using a single noise sample. Similarly, for mini-batch size 4096, two independent noise samples are generated on separate GPUs.

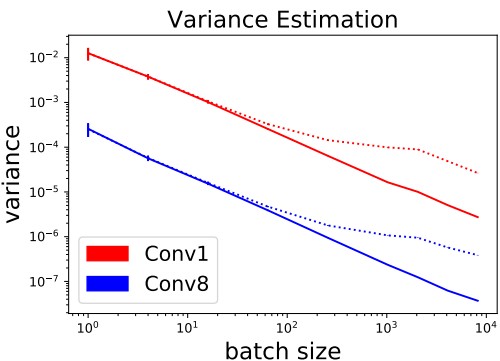

Figure 3: Empirical variance of the gradients when training on multiple GPUs. **Solid:** flipout. **Dotted:** shared perturbations.

## E.2  LARGE BATCH TRAINING WITH FLIPOUT

Fig. 4 shows the training and test error for the large mini-batch experiments described in Section 4.3. For both FC and ConvLe networks, we used the Adam optimizer with learning rate 0.003. We downscaled the KL term by a factor of 10 to achieve higher accuracy.

While Fig. 2a shows that flipout converges faster than shared perturbations, Fig. 4 shows that flipout has the same generalization ability as shared perturbations (the faster convergence doesn't result in overfitting).

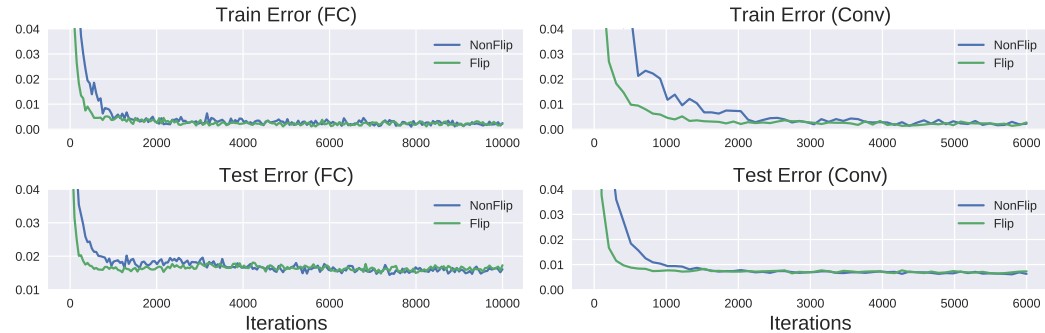

Figure 4: **Left:** The training and test errors obtained by training the FC network on large mini-batches (size 8192) with Bayes by Backprop. **Right:** The training and test errors obtained with ConvLe in the same setting, with mini-batch size 8192.

## E.3  FLIPES V.S. CPUES

Fig. 5 shows that the computational cost of cpuES increases as the model size increases, while FlipES scales better because it runs on the GPU.

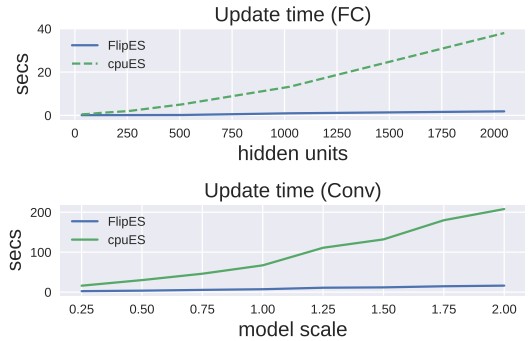

| # Hidden Units | FlipES | cpuES |
|---|---|---|
| 32 | 0.12s | 0.51s |
| 128 | 0.13s | 1.22s |
| 512 | 0.18s | 5.10s |
| 2048 | 1.86s | 38.0s |
| **# Filters** | **FlipES** | **cpuES** |
| 0.25 | 2.3s | 16s |
| 0.75 | 5.48s | 46s |
| 1.0 | 7.12s | 67s |
| 1.5 | 11.77s | 132s |

Figure 5: Per-update time comparison between FlipES and 40-core cpuES (5,000 samples) w.r.t. the model size. We scale the FC network by modifying the number of hidden units, and we scale the Conv network by modifying the number of filters (1.0 stands for 32 filters in the first convolutional layer and 64 filters for the second one).

E.4  LARGE BATCH LSTM TRAINING

The variance reduction offered by flipout allows us to use DropConnect (Wan et al., 2013) efficiently in a large mini-batch setting. Here, we use flipout to implement DropConnect as described in Section 2.1, and use it to regularize an LSTM word-level language model. We used the LSTM architecture proposed by Merity et al. (2017), which has 400-dimensional word embedddings and three layers with hidden dimension 1150. Following Merity et al. (2017), we tied the weights of the embedding layer and the decoder layer. Merity et al. (2017) use DropConnect to regularize the hidden-to-hidden weight matrices, with a single mask shared for all examples in a batch. We used flipout to achieve a different DropConnect mask per example. We applied **WD+Flipout** to both the hidden-to-hidden (h2h) and input-to-hidden (i2h) weight matrices, and compared to the model from Merity et al. (2017), which we call **WD** (for *weight-dropped* LSTM), with DropConnect applied to both h2h and i2h. Both models use embedding dropout 0.1, output dropout 0.4, and have DropConnect probability 0.5 for the i2h and h2h weights. Both models were trained using Adam with learning rate 0.001.

Fig. 6 compares the variance of the gradients of the first-layer hidden-to-hidden weights between WD and WD+Flipout, and shows that flipout achieves significant variance reduction for mini-batch sizes larger than 256. Fig. 7 shows the training curves of both models with batch size 8192. We see that WD+Flipout converges faster than WD, and achieves a lower training perplexity, showcasing the optimization benefits of flipout in large mini-batch settings.

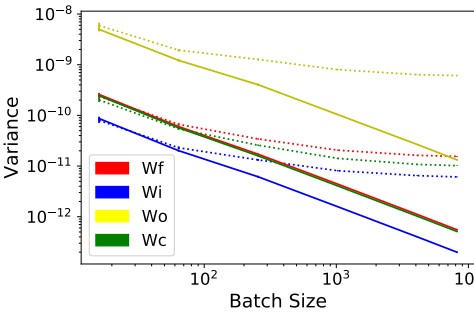

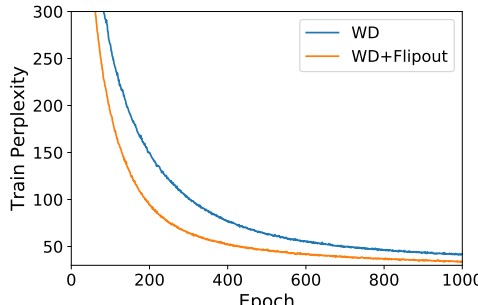

Figure 6: The variance reduction offered by flipout compared to the WD model (Merity et al., 2017). Solid lines represent WD+Flipout, while dotted lines represent WD. The variance is shown for the hidden-to-hidden weight matrices in the first layer: $W_f$, $W_i$, $W_o$, and $W_c$ are the weights for the *forget, input* and *output* gates, and the *candidate cell* update, respectively.

Figure 7: Training curves for WD and WD+Flipout, with batch size 8192.

