# OpenReview forum: "Flipout: Efficient Pseudo-Independent Weight Perturbations on Mini-Batches"
_ICLR.cc/2018/Conference — Accept (Poster)_

### Official Review · AnonReviewer1 · 2017-11-27
**The paper presents a strategy to retain variance reduction while dropping weights rather than activations. It is an important idea that needs more work.**

**Rating:** 6
**Confidence:** 4

**Review:**

The paper is well written. The proposal is explained clearly.
Although the technical contribution of this work is relevant for network learning, several key aspects are yet to be addressed thoroughly, particularly the experiments.

Will there be any values of alpha, beta and gamma where eq(8) and eq(9) are equivalent. In other words, will it be the case that SharedPerturbation(alpha, beta, gamma, N) = Flipout(alpha1, beta1, gamma1, N1) for some choices of alpha, alpha1, beta, beta1, ...? This needs to be analyzed very thoroughly because some experiments seem to imply that Flip and NoFlip are giving same performance (Fig 2(b)).
It seems like small batch with shared perturbation should be similar to large batch with flipout?
Will alpha and gamma depend on the depth of the network? Can we say anything about which networks are better?
It is clear that the perturbations E1 and E2 are to be uniform +/-1. Are there any benefits for choosing non-uniform sampling, and does the computational overhead of sampling them depend on the network depth/size.

The experiments seem to be inconclusive.
Firstly, how would the proposed strategy work on standard vision problems including learning imagenet and cifar datasets (such experiments would put the proposal into perspective compared to dropout and residual net type procedures) ?
Secondly, without confidence intervals (or significance tests of any kind), it is difficult to evaluate the goodness of Flipout vs. baselines, specifically in Figures 2(b,d).
Thirdly, it is known that small batch sizes give better performance guarantees than large ones, and so, what does Figure 1 really imply? (Needs more explanation here, relating back to description of alpha, beta and gamma; see above).

---

> ### Author Response · Authors · 2018-01-05
> **Details of our approach**
>
> Thank you for your careful and insightful feedback.
>
> --> Q: Will there be any values of alpha, beta and gamma where eq(8) and eq(9) are equivalent? Will alpha and gamma depend on the depth of the network? Can we say anything about which networks are better?
>
> Mathematically, eqns. (8) and (9) are equivalent if alpha = 0, and they are nearly identical if beta or gamma dominates. However, we did not observe any examples of either case in any of our experiments. (In fact, beta was indistinguishable from 0 in all of our experiments.) Based on Figure 1, the values seem fairly consistent between very different architectures.
>
> --> Q: FlipES doesn’t outperform NaiveES in Figure 2.
>
> Here, NaiveES refers to fully independent perturbations, rather than a single shared perturbation. Hence, it is an upper bound on how well FlipES should perform, and indeed FlipES achieves this with a much faster wall-clock time (see Fig. 5 in the appendix, cpuES corresponds to noFlip). For clarity, we will rename NaiveES to be IdealES in the final version.
>
> --> Q: Shared perturbation with small batch should be similar to large batch with flipout?
>
> The reason to train with large batches is to take advantage of parallelism. (If we only cared about the number of arithmetic operations, we’d all use batch size 1.) The size of this benefit depends on the hardware, and hardware trends (more GPU cores, TPUs) strongly favor increasing batch sizes. Currently, one may sometimes be able to use small batches to compensate for the inefficiency of shared perturbations, but this is a band-aid which won’t remain competitive much longer.
>
> --> Q: Can we use non-uniform E1, E2 and does the computational overhead of sampling depend on the network depth?
>
> Yes, Proposition 1 certainly allows for non-uniform E1 and E2, although the advantage of this is unclear. In principle, sampling E1 and E2 ought not to be very expensive compared to the matrix multiplications. However, the overhead can be significant if the framework implements it inefficiently; in this case, one can use the trick in Footnote 1.
>
> --> Q: Will the proposed strategy work on standard vision problems including ImageNet and CIFAR?
>
> Our experiments include CIFAR-10, and we see no reason why flipout shouldn’t work on ImageNet. Weight perturbations are not currently widely used in vision tasks, but if that changes, flipout ought to be directly applicable. Our experiments focus on Bayesian neural nets and ES, which inherently require weight perturbations.
>
> Additionally, it was shown that DropConnect (which is a special case of weight perturbation, as we show in Sec. 2.1) regularizes LSTM-based word language models and achieves SOTA on several tasks [1]. Flipout can be directly applied to it, and we show in Appendix E.4 that flipout reduces the stochastic gradient variance compared to [1].
>
> --> Q: Small batch sizes give better performance, what does Fig. 1 imply?
>
> We’re not sure what you mean by this. Due to the variance reduction effects of large batches, one typically uses as large a batch as will fit on the GPU, and sometimes resorts to distributed training in order to use even larger batches. (A batch size of 1 is optimal if you only count the number of iterations, but this isn’t a realistic model, even on a single CPU.)
>
> [1] Merity, Stephen, Keskar, Nitish S., and Socher, Richard. "Regularizing and optimizing LSTM language models." arXiv preprint arXiv:1708.02182 (2017).

---

### Official Review · AnonReviewer3 · 2017-11-27
**Flipout is an important contribution for weight-perturbation algorithms**

**Rating:** 8
**Confidence:** 3

**Review:**

Typical weight perturbation algorithms (as used for e.g. Regularization, Bayesian NN, Evolution
Strategies) suffer from a high variance of the gradient estimates. This is caused
by sharing a weight perturbation by all training examples in a minibatch. More specifically
sharing perturbed weights over samples in a minibtach induces correlations between gradients of each sample, which can
not be resolved by standard averaging. The paper introduces a simple idea, flipout, to
perturb the weights quasi-independently within a minibatch: a base perturbation (shared
by all sample in a minibatch) is multiplied by a random rank-one sign matrix (different
for every sample). Due to its special structure it is possible to vectorize this
per-sample-operation such that only matrix-matrix products (as in the default forward
propagation) are involved. The incurred computational cost is roughly twice as much
as a standard forward propagation path. The paper also proves that this approach
reduces the variance of the gradient estimates (and in practice, flipout should
obtain the ideal variance reduction). In a set of experiments it is demonstrated
that a significant reduction in gradient variance is achieved, resulting
in speedups for training time. Additionally, it is demonstrated that
flipout allows evolution strategies utilizing GPUs.

Overall this is a very nice paper. It clearly lays out the problem, describes
one solution to it and shows both theoretically as well as empirically
that the proposed solution is a feasable one. Given the increasing importance
of Bayesian NN and Evolution Strategies, flipout is an important contribution.

Quality: Overall very well written. Relevant literature is covered and an important
problem of current research in ML is tackled.

Clarity: Ideas/Reasons are clearly presented.

Significance: The presented work is highly significant for practical applicability
of Bayesian NN and Evolution Strategies.

---

> ### Author Response · Authors · 2018-01-05
> **Thank you!**
>
> Thank you for your positive comments and for recognizing the work!

---

### Official Review · AnonReviewer2 · 2017-11-28
**A very pleasant article, but whose actual impact should be made clearer**

**Rating:** 6
**Confidence:** 4

**Review:**

In this article, the authors offer a way to decrease the variance of the gradient estimation in the training of neural networks.
They start in the Introduction and Section 2 by explaining the multiple uses of random connection weights in deep learning and how the computational cost often restricts their use to a single randomly sampled set of weights per minibatch, which results to higher-variance gradient estimatos than could be achieved otherwise. In Section 3 the authors offer to get the benefits of multiple weights without most of the cost, when the distribution of the weights is symmetric and fully factorized, by multiplying sampled-once random perturbations of the weights by a rank-1 random sign matrix. This efficient mechanism is only twice as costly as a single random perturbation, and the authors show how to efficiently parallelize it on GPUs, thereby also allowing GPU-ization of evolution strategies (something so far difficult toachieve). Of note, they provide a theoretical analysis in Section 3.2, proving the actual variance reduction of their efficient pseudo-sampling scheme. In Section 4 they provide quite varied empirical analysis: they confirm their theoretical results on four architectures; they show its use it to regularise on language models; they apply it on large minibatch settings where high variance is a main problem; and on evolution strategies.

While it is a rather simple idea which could be summarised much earlier in the  single equation (3), I really like the thoroughness and the clarity of the exposure of the idea. Too many papers in our community skimp on details and on formalism, and it is a delight to see things exposed so clearly -- even accompanied with a proof.

However, the painful part: while I am convinced by the idea and love its detailed exposure, and the gradient variance reduction is made very clear, the experimental impact in terms of accuracy (or perplexity) is, sadly,  not very convincing. Nowhere in the text did I find a clear rationale of why it is beneficial to reduce the variance of the gradient. The numerical results in Table 1 and Table 2 also do not show a clear improvement: Flipout does not provide the best accuracy. The gain in wall clock could be a factor, but would need to be measured on the figures more clearly. And the validation errors in Figure 2 for Evolution strategies seem to be worse than backprop.The main text itself also only claims performance “comparable to the other methods”.  The only visible gain is on the lower part Figure 2.a on a ConvNet.

This makes me wonder if the authors could do a better job of putting forward the actual advantages of their methods on the end-results: could wall clock measure be put more forward, to justify the extra work? This would, in my mind, strongly improve the case for publication of this article.


A few improvement suggestions:
* Could put earlier more emphasis of superiority to Local Reparameterization Trick in terms of architecture, not wait until Section 2.2 and section 4.1
*Should also put more emphasis on limitations, not wait until 3.1.
* Proposition 1 is quite straightforward, not sure it deserves a proposition, but it’s elegant to put it forward.
* Footnote 1 on re-using the matrices is indeed practical, but also somewhat surprising in terms of bias risks. Could it be explained in more depth, maybe by the random permutations of the minibatches making the bias non systematic and cancelling out?
* Theorem 1: For readability could merge the expectations on the joint distribution as E_{x, \hat \delta W} , rather than separate expectations with the conditional distributions.
* Theorem 1: could the authors provide a clearer intuitive explanation of the \beta term alone, not only as part of \alpha + \beta, especially as it plays such a key role, being the only one that does not disappear? And how do they explain their empirical observation that \beta is close to 0? Any intuition on that?
* Experiments: I salute the authors for providing all the details in exhaustive manner in the Appendix. Very commendable.
* Experiments: I like the empirical verification of the theory. Very neat to see.

Minor typo:
* page 2 last paragraph, “evolution strategies” is plural but the verbs are used in singular (“is black box”, “It doesn’t”, “generates”)

---

> ### Author Response · Authors · 2018-01-05
> **Flipout offers optimization benefits for large batch sizes**
>
> Thank you for your careful and insightful feedback.
>
> -> Q: Why is it beneficial to reduce the variance of the gradient, flipout doesn’t provide the best accuracy, wall-clock time advantage?
>
> The importance of variance reduction in SGD is well-established. Variance reduction is the whole reason for using batch sizes larger than 1, and some well-known works [1] have found that with careful implementation, the variance-reducing effect of large batches translates into a linear optimization speedup. Whether this relationship holds in a particular case depends on a whole host of configuration details which are orthogonal to our paper; e.g. the aforementioned paper had to choose careful initializations and learning rate schedules in order to achieve it. Still, we can confidently say there is no hope for unlocking the optimization benefits of large batches unless one uses some scheme (such as flipout) that enables variance reduction.
>
> Note that we do in fact observe significant optimization benefits (flipout converges 3X faster in terms of iterations), as shown in Figure 2(a). Additionally, although flipout is 2X more computationally expensive in theory, it can be implemented more efficiently in practice. For example, we can send each of the two matmul calls to a separate TPU chip, so they are done in parallel. Communication will add overhead but it shouldn’t be much, as it is only two chips communicating and the messages are matrices of size [batch_size, hidden_size] rather than the set of full weights.
>
> With respect to the advantages offered by flipout for training LSTMs, we have conducted new experiments to compare regularization methods, and have updated Section 4.2 and the results in Table 2. We found that using flipout to implement DropConnect for recurrent regularization yields strong results, and significantly outperforms the other methods in both validation and test perplexity. For our original word-level LSTM experiments, we used the setup of [2], with a fixed learning schedule that decays the learning rate by a factor of 1.2 each epoch starting after epoch 6. In our new experiments, we decay the learning rate by a factor of 4 based on the nonmonotonic criterion introduced in [3]; the perplexities of all methods except the unregularized LSTM are reduced compared to the previous experiments. Using flipout to implement DropConnect allows us to use a different DropConnect mask per example in a batch efficiently (compared to [3], which shares the weights between all examples).
>
> We also added Appendix E.4, which shows that using flipout with DropConnect yields significant variance reduction and faster training compared to using a shared DropConnect mask for all examples (as is done in [3]).
>
> -> Q: ES seems to be worse than backprop.
>
> We’re not advocating for ES to replace backprop. The main comparison in this section is between NaiveES and FlipES; we show that FlipES behaves like NaiveES, but is more efficient due to parallelism. Our reason for including the backprop comparison is to show that this is an interesting regime to investigate. One might have thought that ES would hopelessly underperform backprop (since the latter uses gradients), but in fact FlipES turns out to be competitive. The reason this result is interesting is that unlike backprop, ES can also be applied to non-differentiable models.
>
> -> Q: Footnote 1 with bias risk.
>
> The trick in Footnote 1 does not introduce any bias. Proposition 1 implies that the gradients are unbiased for any distribution over E which is independent of Delta W. This applies in particular to deterministic E (which is trivially independent), so E can be fixed throughout training. Such a scheme may not achieve the full variance reduction, but it is at least unbiased. Note that we do not use this trick in our experiments.
>
> -> Q: Why is it close to 0 in practice, intuitive explanation of beta term?
>
> Beta is the estimation variance when E is marginalized out. We’d expect this term to be much smaller than the full variance because it’s marginalizing over a symmetric perturbation distribution, so the perturbations in opposite directions should cancel. The finding that it was so close to zero was a pleasant surprise.
>
> We also thank the reviewer for the suggestions for improvement. We will revise the final version to take them into account.
>
>
> [1] Goyal, Priya, et al. "Accurate, large minibatch SGD: Training ImageNet in 1 hour." arXiv preprint arXiv:1706.02677 (2017)
> [2] Yarin Gal and Zoubin Ghahramani.   A theoretically grounded application of dropout in recurrent
> neural networks. In Advances in Neural Information Processing Systems (NIPS), pp. 1019–1027, (2016).
> [3] Merity, Stephen, Keskar, Nitish S., and Socher, Richard. "Regularizing and optimizing LSTM language models." arXiv preprint arXiv:1708.02182 (2017).

---

> > ### Comment · AnonReviewer2 · 2018-01-12
> > **Reviewer's Answer to Authors' Response**
> >
> > I thank the authors for their response. I am disappointed that their revised paper does not provide any further explanation of why reducing the variance of SGD matters. The explanation from the authors in their response is that:
> > * "it is well established": citing only one paper is not convincing, and even if it were so clearly well-established, your duty to your readers is to make the article's motivation as self-contained as possible.
> > * "that's why we use minibatches larger than 1" : there's a world of diminishing returns between "larger than 1", which is indeed extreme, and "1000+"
> > * "it is an orthogonal problem": I beg to differ. Knowing the setup in which your method can help is very much aligned with developing the method.
> >
> > As such, after revision, it is with regret that I maintain my rating of "6: Marginally above acceptance threshold".

---

> > > ### Author Response · Authors · 2018-01-13
> > > **Response to Reviewer 2**
> > >
> > > Thank you for your comment.
> > >
> > > Variance reduction is a central issue in stochastic optimization, and countless papers have tried to address it. To summarize, lower variance enables faster convergence and hence improves the sample efficiency. We gave one reference above, but there are many more that we did not mention (a few more examples are [1-12]). Furthermore, the variance of the stochastic gradients is arguably the most serious problem facing policy gradient methods such as evolution strategies, and some fundamental algorithms like REINFORCE are essentially variance reduction methods. So hopefully the importance of variance reduction is clear.
> > >
> > > We demonstrated consistently large variance reduction effects, and showed that in at least some cases, this leads to more efficient training (see Figure 2.a).  In addition: 1) we show in Table 2 that flipout applied to DropConnect outperforms all the other methods by a significant margin (73.02 test perplexity, compared to 75.31 for the next-best method); and 2) we show in Appendix E.4 that flipout applied to DropConnect converges faster than DropConnect with shared masks (which is currently the SOTA method).
> > >
> > > Regarding when our method helps: most straightforwardly, it helps when SGD is suffering from high estimation variance. This turned out to be the case for some of the BNNs we experimented with, as well as for ES (which is notoriously high-variance). As we’ve pointed out, estimation variance will become a more serious bottleneck as hardware trends favor increasing batch sizes. These factors give simple rules of thumb for when flipout will be useful.
> > >
> > > [1] Andrew C. Miller et al. Reducing reparameterization gradient variance. In NIPS, 2017.
> > > [2] Alberto Bietti and Julien Mairal. Stochastic optimization with variance reduction for infinite datasets with finite sum structure. In NIPS, 2017.
> > > [3] Sashank J. Reddi et al. Stochastic variance reduction for nonconvex optimization. In ICML, 2016.
> > > [4] Soham De, Gavin Taylor, and Tom Goldstein. Variance reduction for distributed stochastic gradient descent. arXiv preprint arXiv:1512.01708, 2015.
> > > [5] Aaron Defazio, Francis Bach, and Simon Lacoste-Julien. Saga: A fast incremental gradient method with support for non-strongly convex composite objectives. In NIPS, 2014.
> > > [6] Aaron Defazio et al. Finito: A faster, permutable incremental gradient method for big data problems. In ICML, 2014.
> > > [7] Reza Harikandeh et al. Stop wasting my gradients: Practical SVRG. In NIPS, 2015.
> > > [8] Rie Johnson and Tong Zhang. Accelerating stochastic gradient descent using predictive variance reduction. In NIPS, 2013.
> > > [9] Sashank J. Reddi et al. On variance reduction in stochastic gradient descent and its asynchronous variants. In NIPS, 2015.
> > > [10] Nicolas L Roux, Mark Schmidt, and Francis R Bach. A stochastic gradient method with an exponential convergence rate for finite training sets. In NIPS, 2012.
> > > [11] Chong Wang, Xi Chen, Alex J Smola, and Eric P Xing. Variance reduction for stochastic gradient optimization. In NIPS, 2013.
> > > [12] Lin Xiao and Tong Zhang. A proximal stochastic gradient method with progressive variance reduction. SIAM Journal on Optimization, 24(4):2057–2075, 2014.

---

### Author Response · Authors · 2018-01-05
**Updated paper**

We thank the reviewers for their helpful comments.

We updated the regularization experiments in Section 4.2, including the results in Table 2. We show that flipout applied to DropConnect outperforms all the other methods we investigated.

We also added Appendix E.4, in which we show that for large-batch LSTM training: 1) using DropConnect with flipout achieves significant variance reduction compared to using a shared DropConnect mask for all examples; and 2) DropConnect with flipout converges faster than DropConnect with shared masks, showcasing the optimization benefits of using flipout.

---

### Decision · Program_Chairs · 2018-01-29
**ICLR 2018 Conference Acceptance Decision**

**Decision:**

Accept (Poster)

**Comment:**

Thank you for submitting you paper to ICLR. The idea is simple, but easy to implement and effective. The paper examines the performance fairly thoroughly across a number of different scenarios showing that the method consistently reduces variance. How this translates into final performance is complex of course, but faster convergence is demonstrated and the revised experiments in table 2 show that it can lead to improvements in accuracy.